

**Pore-water in marine sediments associated to gas hydrate dissociation**
**offshore Lebu, Chile.**
Carolina Cárcamo[1,2], Iván Vargas-Cordero[1], Francisco Fernandoy[1], Umberta
Tinivella[3], Diego López-Acevedo[4], Joaquim P. Bento[5], Lucía Villar-Muñoz[6], Nicole
Foucher[1], Marion San Juan[1], Alessandra Rivero[1]
[1] Universidad Andres Bello, Facultad de Ingeniería, Quillota 980, Viña del Mar, Chile
[2] Centro de Investigación Marina Quintay. CIMARQ. Facultad de Ciencias de la Vida. Universidad
Andres Bello, Viña del Mar, Chile.
[3] OGS Istituto Nazionale di Oceanografia e di Geofisica Sperimentale, Borgo Grotta Gigante 42/C,
34010, Sgonico, Italy.
[4] Universidad de Concepción, Departamento de Oceanografía, Programa COPAS Sur-Austral,
Campus Concepción Víctor Lamas 1290, P.O. Box 160-C, Concepción, Chile
[5] Escuela de Ciencias del Mar, Pontificia Universidad Católica de Valparaíso, Av. Altamirano 1480,
2360007 Valparaíso, Chile.
[6] GEOMAR Helmholtz Centre for Ocean Research, Wischhofstr. 1-3, 24148 Kiel, Germany.
**ABSTRACT**
Gas hydrate occurrences along the Chilean margin has been documented, but the
processes associated to fluid escapes originated by gas hydrate dissociation yet are
unknown. Here, we report morphologies growing related to fluid migration in the
continental shelf offshore western Lebu (37 °S) by analysing mainly geochemical
features. In this study oxygen and deuterium stable water isotopes in pore water
were measured. Knowledge was completed by analysing bathymetric data,
biological and sedimentological data. From bathymetric interpretation a positive
relief at 127 m below sea level was recognised; it is oriented N55°E and
characterised by five peaks. Moreover, enrichment values for $\delta^{18}O$ (from 0.0 to
1.8‰) and δD (from 0.0 to 5.6‰) were obtained. These are typical values related to
hydrate melting during coring and post-sampling. The evident orientation of positive
relief could be associated with faults and fractures reported by others authors, in
which these structures constitute pathways for fluid migration from deep to shallow
zones. Finally, benthic foraminifera observed in the core sample can be associated
to seep areas. On the basis of theoretical modelling, we conclude that the positive
relief correspond to mud growing processes related to gas hydrates dissociation and
represent a key area to investigate fluid migration processes.
**Keywords**: gas hydrate dissociation, stable isotopes, pore water, mud growing, fluid
migration



## 1. Introduction

Morphological features associated with fluid escapes along continental margins (e.g.: mud volcanoes, mud mounds, pockmarks, seeps) have been worldwide reported (Van Rensbergen et al., 2002; Loncke et al., 2004; Hovland et al., 2005; Lykousis et al., 2009; Chen et al., 2010). Fluid escapes can be formed mainly by biogenic and thermogenic methane gas and water. The gas can give place to gas hydrate formation in marine sediments if pressure and temperature conditions are adequate (Sloan, 1998), in which the gas is trapped in a lattice of water molecules. Gas hydrate occurrences along the Chilean margin are distributed from 33 to 57°S (Bangs et al., 1993; Froelich et al., 1995; Morales, 2003; Grevemeyer et al., 2003; Rodrigo et al., 2009; Vargas-Cordero et al., 2010, 2010a, 2016, 2017; Villar-Muñoz et al., 2014, 2018). Several studies have documented fluid escapes related to gas hydrate dissociation through faults and fractures (e.g., Yin et al., 2003; Thatcher et al., 2013). Among others, common techniques often used to recognize such processes are: biological, geochemical and geophysical analyses. Biological indicators as benthic foraminifera, bivalve shells and microbial communities have been related with fluid escapes (Reed et al., 2002; Chen et al., 2007; Karstens et al., 2018). Moreover, enriched stable water isotope values have been measured from pore water of marine sediments. Hesse (2003) and Kvenvolden and Kastner (1990) reported in extensive articles several cases of enriched stable water isotope values from different regions, including the Chilean coast. Finally, geophysical studies have allowed identifying morphologies associated with fluid escapes by using bathymetric, backscatter and high resolution images (Sager et al., 2003; Loncke et al., 2004; Tinivella et al., 2007). Well and seismic data interpretations have allowed recognizing active structural domain offshore Arauco basin (Melnick and Echtler, 2006; Melnick, 2006a). During the depositional history of Arauco Basin, numerous tectonic phases have been recognized, including subsidence and uplift episodes that gave place to accretion and erosion of the prism (Bangs and Cande, 1997; Lohrmann, 2002).Cretaceous-Plio-Pleistocene marine and continental sequences configure a cyclic sedimentary complex. Sedimentary sequences are composed by alternating of marine and continental deposits. From base to top, these are: Quiriquina (Biró-Bagóczky, 1982), Pilpilco, Curanilahue, Boca Lebu, Trihueco, Millongue, Ranquil, Tubul and Arauco formations (Pineda, 1983; Viyetes et al., 1993; Muñoz-Cristi, 1956; Muñoz-Cristi, 1968). Nahuelbuta Range is composed by Carboniferous-Permian granitoides (Coastal Batholith) intruding the Paleozoic-Mesozoic metamorphic rocks. Moreover, gas and carbon reservoirs have been identified along the Arauco basin (Mordojovich, 1974; González, 1989).

This study aims at characterizing an identified positive relief in order to understand its origin by using geochemical, sedimentological and bathymetric data. The study area is located on continental shelf close to 150 m below sea level (mbsl) and includes part of Arauco basin (Fig. 1).

## 2. Data and Methods

### 2.1 Data



In the framework of project entitled "Identification and quantification of gas emanations associated with gas hydrates (FONDECYT 11140214)" sedimentological, geochemical and bathymetric study offshore Lebu was performed (Fig. 1). In 2016 and 2017 two marine campaigns onboard R/V Kay Kay II were carried out collecting bathymetric data, seawater samples and marine sediments Marine sediment samples were recovered by using gravity corer (diameter of 9 cm) at around 127 mbsl and the gravity corer drilled as deep as 240 cm into marine sediments (GC-02). The core collected is located around positive relief close to 73°44'25"W-37°36'10"S (Fig. 2). The GC-02 core was divided into four sections of 60 cm long (S01, S02, S03, S04); each section one was frozen and analysed at Sedimentology Andres Bello University's laboratory (Viña del Mar, Chile)

The water samples were collected by Niskin bottles at 5 depths (0 m, 10 m, 20 m, 50 m and bottom); temperature, conductivity, dissolved oxygen and pH was determined with multiparameter measurer model IP67. These parameters were measured at the two ends of the identified lineament, i.e., the first station located to the south and the second one to the north (Fig. 2).

**2.2 Methods**

The procedure includes: a) bathymetric data processing and b) sedimentological, physical-chemical and geochemical analyses of seawater and marine sediment samples.

Bathymetric and sound velocity data were acquired by using multihaz Reson SeaBat 7125 echosounder (400 kHz, 0.5° x 1°) and SVP90 probe, respectively. Besides, an AML Oceanographic Model Minos X sound velocity profiler was used. A preliminary processing was performed onboard by using PDS2000 commercial software. This software allows correcting bathymetric data in real time by using SVP90 and AML information and ship motions (pitch, roll, yaw and heave). The bathymetric data processing was performed by using open-source MB-System software (Caress and Chayes, 2017). In this step, bathymetric data were converted in MB-System format in order to attenuate tide and scattering effects. In the first step, bathymetric grids with nearneighbor interpolation algorithm were created by using open-source software Generic Mapping Tools (GMT, Wessel et al., 2013). The algorithm builds cell values in depth rectangular distributed, in which each node value corresponds to the weighted average of around probes of search circle of 1 arc second. Besides the selected grid was configured with spatial resolution of 0.2 arc seconds. Finally, a median filter of 10 m width was applied in order to smooth the grid.

Grain size analysis includes sieving method where sediments pass through (by agitation) meshes; in our case, 50 g of sediment sample were sieved by using the following mesh sizes: 60, 80, 120 and 230. The pipette method was adopted in order to separate clay and silt fractions by selecting 15 g of mud sample. Statistical parameters were calculated in agreement with reported formulas (Folk and Ward, 1957; Carver, 1971; Scasso and Limarino, 1997).

Seawater physical-chemical properties (temperature, pH, salinity and dissolved
oxygen) in proximity of the positive relief were obtained by using the multiparameter
Meter (IP67, model 8602). The multiparameter Meter has different types of probes
or electrodes, which must be selected according to the required function and to
obtain accurate measurements. Temperature was measured in Celsius degree, with
an accuracy of +/0.5°C, while pH was directly related to the ratio of the
concentrations of hydrogen ions [H +] and hydroxyl [OH] (Cabo, 1978) with an
accuracy of +/-0.1. Salinity was obtained from the conductivity, which depends on
the number of dissolved ions per unit volume and the mobility of the ions; the
accuracy is +/-0.1. Finally, dissolved oxygen can be measured both in % and in mg/L,
with accuracy +/-3%; in our case, it was expressed in %.
The core was cut in sections of 10 cm long and then the main physical-chemical
parameters were measured including pore water (w%), porosity (Φ), the content of
solid material per unit volume, expressed as apparent density (ρ; Salamanca and
Jara, 2003) and total organic matter (TOC). Finally, samples were dried in forced air
oven at 60°C for 36 hours and in a desiccator for 30 minutes.
TOC content was measured by gravimetric determination of weight loss through
loss-on-ignition method (Byers et al., 1978; Luczak et al., 1997). In our case, 2 g of
dry sediment sample was calcined in muffle at 500 °C for 5 hours and, then, it was
placed in desiccator for 30 minutes until to register constant weight in order to reduce
the associated error. Pore water from core was extracted using an ACME lysimeter
(0.2 µm) in order to analyse oxygen and deuterium stable water isotopes. The pore
water extraction procedure includes: a) corer cutting in sections of 5 cm long, b)
centrifugation, c) pore water extraction by using Rhizon MOM with pore sizes ranging
to 0.12 to 0.18 µm and d) stable water isotope determination by Cavity Ring Down
Spectroscopy (CRDS) method at the Laboratorio de Análisis Isotópico (LAI) at the
Universidad Andrés Bello (Viña del Mar, Chile).
Oxygen and deuterium water isotope analyses were evaluated using LIMS (Coplen
and Wassenaar, 2015) and normalized to the VMSOW-SLAP scale and reported as
δ-values for oxygen ($\delta^{18}O$) and deuterium ($\delta D$).

## 3. Results

From bathymetric data, a positive relief located at 127 mbsl with orientation N55°E
was recognised. The relief shows an elevation of about 6 m above the seafloor, an
extension of 700 m length and a width of 50 m (Fig. 2). Five peaks along the relief
were observed.
Grain size analysis shows a constant values in depth. The average grain size value
corresponds to sandy mud textural group. Silt size reaches 60% of total volume (Fig.
3). Physical-chemical parameter distribution of core GC-02 are detailed in Table 1.
A slightly variation of water content  (average equal to 43.1%), apparent density
(average equal to 1.6 g/cm³) and porosity (average equal to 66.9%) were detected.
TOC values show a variable trend with maximum value equal to 8.7% of total volume





located at 2.2 m, while the minimum value is equal to 5.1% of total volume detected
at 0.4 m (Fig. 4).
Pore stable water isotope analysis of marine sediment core shows positive values
ranging from 0.0 up to +1.8‰ for of $\delta^{18}O$ and 5.6‰ for δD, respectively (Fig 5).
Stable water isotope δ-values show a positive trend (enrichment) towards the bottom
of the sediment core, with values close to 0 at the top in the sediment-water interface,
and a restricted variability for all samples analysed (Std. Dev. 0.33 and 0.95 for $\delta^{18}O$
and δD, respectively). It should be noticed that no negative values were found along
the core.
Benthic foraminiferal accumulations in shallow level of core (0-60 cm) showing
globose and elongated morphologies. The following genders of opportunistic
foraminifera were recognized: *Globobulimina, Bolivina, Valvulineria, Anomalinoides,*
*Uvigerina, Oridorsalis and Quinqueloculina* (Fig. 6).
Temperature values range from 12 to 14 °C in seawater samples, registering
maximum values in correspondence of shallow levels, while minimum values were
found in deep levels. Salinity and dissolved oxygen values show a similar trend with
maximum values equal to 33‰ and 60% located at 20 mbsl, respectively. Minimum
values of salinity (31 %) and dissolved oxygen (66.2%) were measured in station 1
at 0.6 mbsl, respectively. pH values range from 7.5 to 8.1 (Fig. 5).
**4. Discussion and conclusion**
The stable water isotope composition of pore water represents a strong evidence of
gas-hydrate dissociation. Figure 5a shows the stable water isotope profile of the
entire core, showing an evident trend with values close to 0‰ at water-sediment
interface to positive values at the bottom of the core (~2‰ and 6‰ for $\delta^{18}O$ and δD,
respectively). This trend shows the influence of sea-water mixing on the top and a
different source at the bottom of the core. Positive values of meteoric waters are
mostly associated to high evaporation rates, which could be discarded in the context
of this investigation. Positive $\delta^{18}O$ values have been reported for clay minerals
dewatering; however in this case a δD depletion rather than enrichment is expected
(Hesse, 2003). Nonetheless, the co-isotope relationship (Fig. 5b) of our samples
shows that pore waters stable water composition have a positive correlation (i.e.:
simultaneous enrichment of $\delta^{18}O$ and δD). Additionally, the meteoric origin of the
pore water can be rejected as shown in Fig. 5c, as pore waters fall away from the
Global Meteoric Water Line (GMWL), which defines the fractionation processes
during the hydrological cycle (Craig, 1961). Stable water isotope enrichment of pore
has been related to hydrate melting during coring and post-sampling (Hesse, 2003;
Tomaru et al., 2006), which are preferentially enriched by heavy stable water
isotope.
The infaunal foraminifera, found in the shallower sediment sample (e.g Bolivina and
Uvigerina), could be associated with modern cold seep, since they can be
metabolising seeping methane, directly or indirectly exploiting the available
geochemical energy source (Jones, 2014). Besides, benthic foraminifera are



associated with high organic content ambient, low oxygen conditions and cold seep
occurrences (Hill et al, 2003; Rathburn et al. 2000).
In the study area across the continental slope zone, gas phases concentrations were
estimated by Vargas et al. (2010a), reporting 15% of total volume for hydrates and
0.2% of total volume for free gas. Several studies argue that lateral fluid migration
can occur from deep levels through faults and fractures canalising fluids and giving
place to mud mounds and mud volcanoes (Yin et al., 2003; Thatcher et al., 2013).
Other studies in our study area have reported faults extending wards offshore zones
(Melnick et al., 2009; Vargas-et al., 2011; Becerra et al., 2013). Moreover, gas
accumulations can reach shallow areas because the base of gas hydrate stability
zone (GHSZ) can be very shallow in the continental shelf, as indicated by theoretical
modelling. In fact, in order to understand where the gas hydrate is stable versus
seawater depth, the theoretical base of the GHSZ was calculated assuming a
geothermal gradient of 30 km/°C (in agreement with Vargas-Cordero et al., 2010a)
and a mixture of 95% of methane and 5% of ethane (in agreement with measures at
ODP Site 1235). Details about the method are reported in Vargas-Cordero et al.
(2017). Note that seismic data acquired in our study area detected the presence of
the hydrate and the free gas, confirming that this area is characterized by relevant
upward fluid flow (Vargas-Cordero et al., 2010a). As shown in Fig. 7, the theoretical
base of GHSZ reaches the seafloor at a seawater depth of about 400 m; so, at
shallower seawater depth the hydrate is not stable and only free gas can be present.
Note that in our study area the continental shelf is shown narrow (15 km width)
favouring that fluids associated to gas hydrate dissociation and gas accumulations
can migrate wards shallow areas from the base of GHSZ. It is important to notice
that in other areas at high latitudes, an extent reduction of the GHSZ, was observed
due to the warming over the last 20000 years (i.e., Westbrook et al., 2009; Thatcher
et al., 2013). To verify a similar trend in our study area, we modelled the theoretical
base of the GHSZ supposing past temperature conditions reported by paleoclimatic
reconstruction studies (Kim et al., 2002; Lamy and Kayser, 2009), i.e. a decrease of
the seawater bottom temperature of 1 °C, 2 °C, 3 °C, 4 °C, and 5°C (Fig. 7). The
modelling indicates that the origin of the mud structures analysed in this paper can
be related to hydrate dissociation caused by the increase of seawater bottom
temperature in the past.
Grain size results can be associated with flow hydrodynamic conditions, in which
mud and sand could be related with coastal and beach systems, fluvial or deltaic
deposits (Mordojovich, 1981). Slightly vertical variations allow us to define a
relationship between physical-chemical parameters (W, Φ, ρ and MOT) and grain
sizes results. Studies reported by Pineda (2009) argue that clay and silt presence in
marine sediments are capable to retain organic wastes increasing TOC values. The
values ranging from 0.5 to 10% reported by Pineda (2009) are in agreement with
values presented in this study.
The results of the seawater analysis show typical values of temperature, salinity,
dissolved oxygen and pH, which are associated with seawater masses. The
temperature in the seawater column increases in shallow levels, whereas it





decreases in deep levels. An opposite trend regarding salinity and dissolved oxygen
values were recognized; in effect, when the oxygen solubility decreases, the
temperature and salinity increases (Cabo, 1978). The pH values ranging from 7.4 to
8.4 can be associated with seawater alkalinity. The highest values are often detected
on the seawater surface (Cabo, 1978).  No relationships between seawater physical-
chemical parameters and our conclusion were found, which can be explained due
to: a) discrete data collected (e.g five seawater samples were collected in a column
of 130 m) or b) upwelling and downwelling processes reported in this area (Parada
et al., 2012) could give place to water mass exchange preventing to observe
significant variations.
We can conclude that the positive relief can be associated with mud mound growing
by fluid flux supply canalised by faults and fractures. These fluids probably are
related to gas hydrates dissociation, in which gas and water migrate from deeper to
shallower areas.

## Acknowledgements

We are grateful to CONICYT (Fondecyt de Iniciación N°11140216), which partially
supported this work. The authors are grateful to Michela Giustiniani for constructive
discussions and useful comments. Special thanks to Mauricio and Daniel from the
palaeontology laboratory (UNAB - Viña del Mar), who helped us with the foraminifera
identification.

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





**Figures**

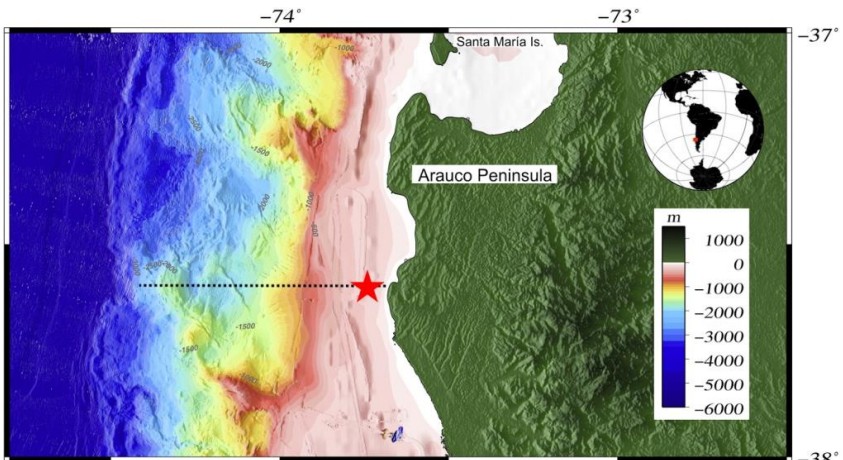


**Figure 1**: Location map of the studied area. Red star shows core recovery and
bathymetric survey. Dashed line shows the bathymetric profile used in Fig. 7.

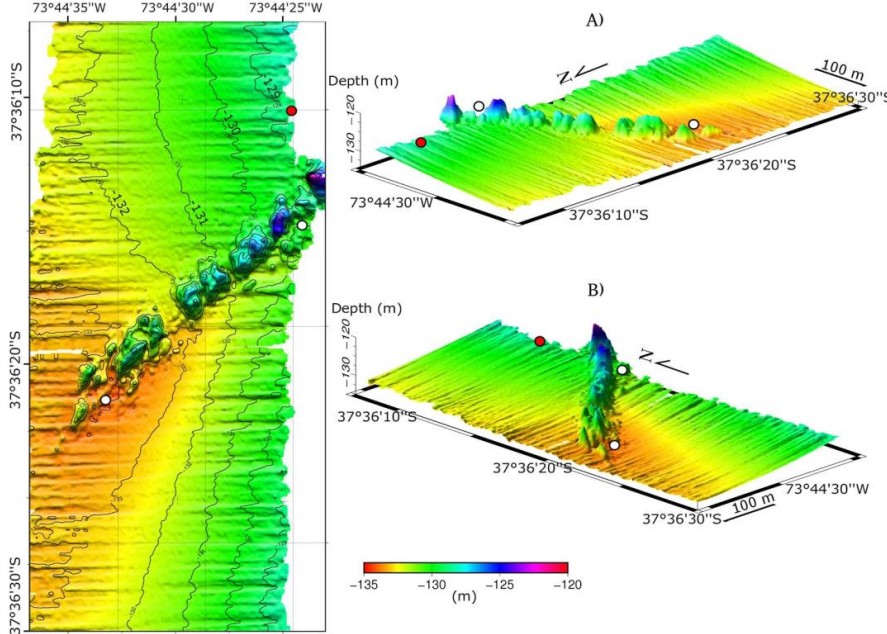


**Figure 2**: Bathymetric map indicating location core GC-02 (red circle). In A) and B)
3D images with orientation NW and SW respectively. The white circles indicate the
position of the two water samples.



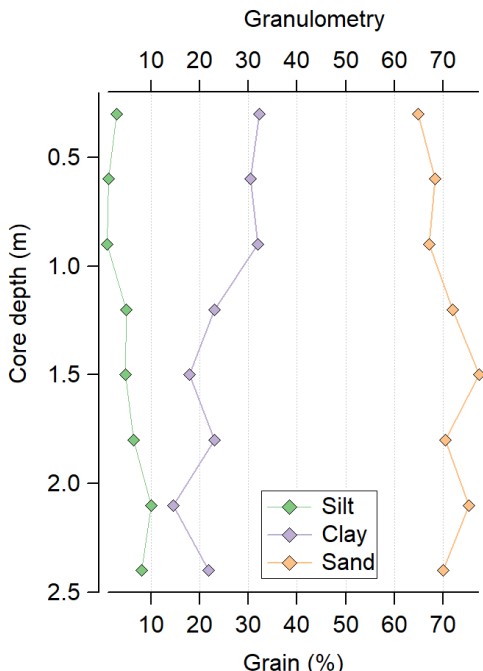


**Figure 3**: Grain size distribution in marine sediments (core GC02).

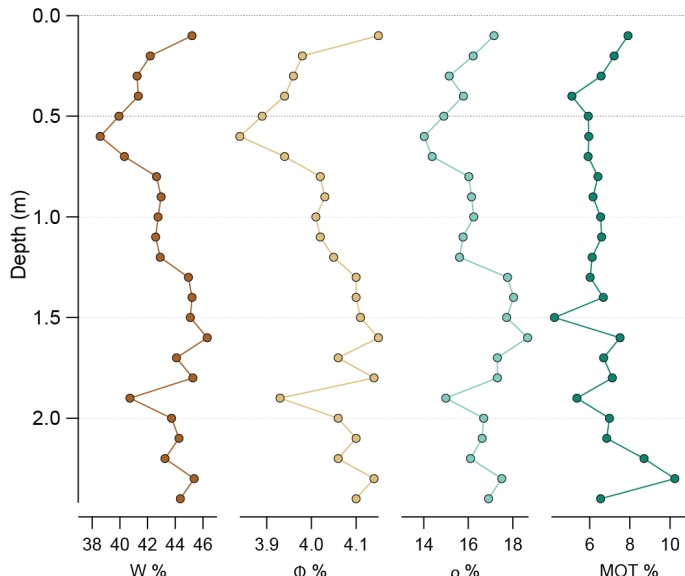


**Figure 4**: Physical-chemical parameters distribution in marine sediments (core GC02).

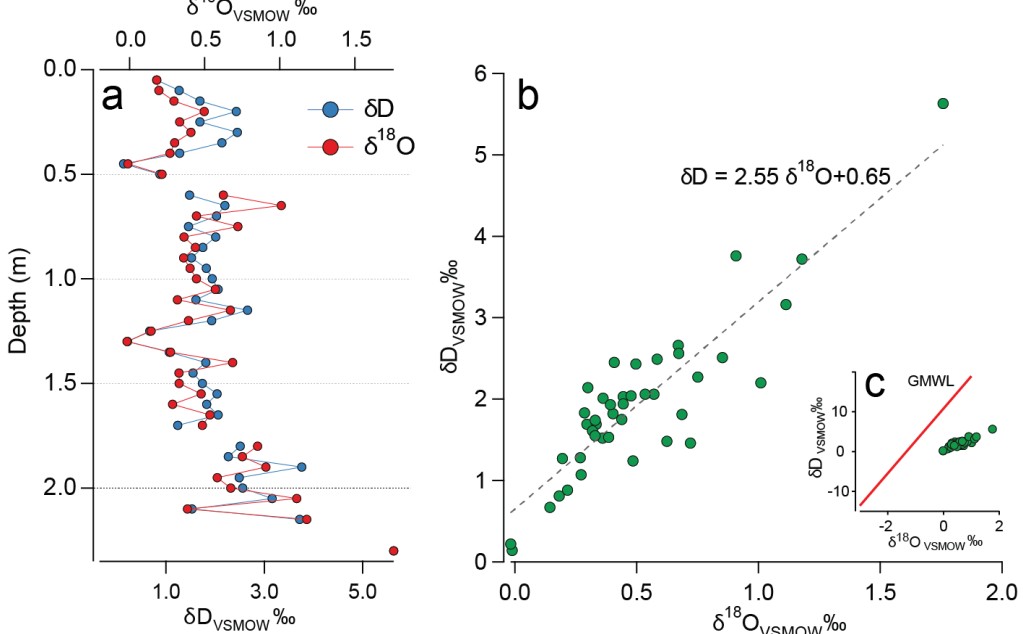

**Figure 5**. Oxygen (δ18O) and deuterium (δD) stable water isotope distribution in
sediment from: a. Depth profile of the core; b. co-isotope distribution of the pore
water and c. relationship of the co-isotope distribution of pore water samples against
the global meteoric water (GMWL)






**Figure 6**: Benthic foraminifera. In (Fig. 1a) *Globobulimina*, lateral view (10x); (Fig. 1b) *Globobulimina*, lateral view (10x); (Fig. 2) *Bolivina*, lateral view (5x); (Fig. 3a) *Valvulineria*, lateral view (5x); (Fig. 3b) *Anomalinoides*, lateral view (5x); (Fig. 4) *Uvigerina*, lateral view (5x); (Fig. 5a) *Oridorsalis*, lateral view (5x); (Fig. 5b) O*ridorsalis*, lateral view (5x); (Fig. 6) *Quinqueloculina*, lateral view (10x).





545

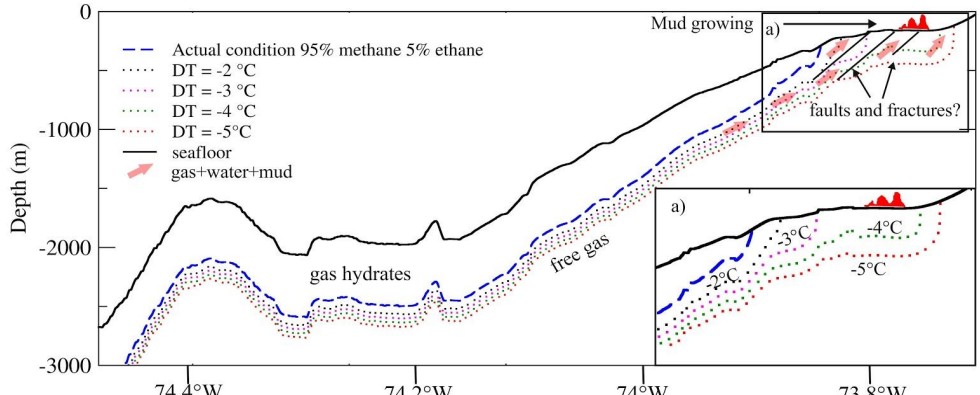

546

**Figure 7**: Schematic profile explaining mud growing formation (in red). The profile location is shown in Fig. 1. Dashed lines show theoretical bases of GHSZ by using geothermal gradient of 30°C/km for several scenarios supposing that the hydrate is formed by a mixture of 95% of methane and 5% of ethane. The blue dashed line indicates the actual theoretical base of the GHSZ. The dotted lines indicate the theoretical base of GHSZ supposing a decrease of the bottom temperature of 2 °C (black dotted line), 3 °C (magenta dotted line), 4 °C (green dotted line) and 5 °C (red dotted line). The black solid line indicates the seafloor. The pink arrows indicate the direction of the fluid/mud outflow. Possible faults and fractures are also reported as black lines.

















**Table 1**: Physical-chemical parameter distribution in marine sediments

| Depth (m) | W (%) | φ (%) | ρ (g/cm³) | MOT (%) |
|---|---|---|---|---|
| 0.1 | 45.2 | 68.8 | 1.6 | 7.9 |
| 0.2 | 42.2 | 66.1 | 1.6 | 7.2 |
| 0.3 | 41.2 | 65.2 | 1.6 | 6.6 |
| 0.4 | 41.3 | 65.3 | 1.6 | 5.1 |
| 0.5 | 39.9 | 64.0 | 1.6 | 5.9 |
| 0.6 | 38.6 | 62.7 | 1.7 | 6.0 |
| 0.7 | 40.3 | 64.4 | 1.6 | 5.9 |
| 0.8 | 42.7 | 66.5 | 1.6 | 6.4 |
| 0.9 | 43.0 | 66.8 | 1.6 | 6.2 |
| 1 | 42.8 | 66.6 | 1.6 | 6.5 |
| 1.1 | 42.6 | 66.5 | 1.6 | 6.6 |
| 1.2 | 42.9 | 66.7 | 1.6 | 6.1 |
| 1.3 | 45.0 | 68.6 | 1.6 | 6.0 |
| 1.4 | 45.2 | 68.8 | 1.6 | 6.7 |
| 1.5 | 45.1 | 68.7 | 1.6 | 4.2 |
| 1.6 | 46.3 | 69.7 | 1.5 | 7.5 |
| 1.7 | 44.1 | 67.8 | 1.6 | 6.7 |
| 1.8 | 45.3 | 68.8 | 1.6 | 7.1 |
| 1.9 | 40.7 | 64.7 | 1.6 | 5.4 |
| 2 | 43.7 | 67.5 | 1.6 | 7.0 |
| 2.1 | 44.3 | 68.0 | 1.6 | 6.8 |
| 2.2 | 43.3 | 67.1 | 1.6 | 8.7 |
| 2.3 | 45.4 | 68.9 | 1.6 | 6.9 |
| 2.4 | 44.4 | 68.0 | 1.6 | 6.5 |
| **Average** | **43.1** | **66.9** | **1.6** | **6.5** |
| **Minimum** | **38.6** | **62.7** | **1.5** | **4.2** |
| **Maximum** | **46.3** | **69.7** | **1.7** | **8.7** |