# Peer review of "Pore-water in marine sediments associated to gas hydrate dissociation offshore Lebu, Chile."

_Hydrology and Earth System Sciences, 2018_

## Referee Comment (RC1) · Anonymous Referee #1 · 14 Nov 2018

Carcamo et al in the manuscript "Pore-water in marine sediments associated to gas hydrate dissociation offshore Lebu, Chile" use bathymetric, sedimentological, foraminiferal and isotopic measurements to identify and explain the formation of a positive relief along the Chilean margin. The combination of the different approaches – including theoretical modeling - leads the authors to conclude that the positive relief identified is the result of mud growing processes associated with gas hydrate dissociation in a specific region where cold seeps occur in previously identified faults and fractures.

General comments

As a regular reader of HESS papers, I feel like the paper by Carcamo et al does not fall into the scope of HESS journal. Although occurring at a margin which is by definition at

the interface between continents and oceans, the scope of the paper is in my opinion more related to marine sedimentology and/or marine geochemistry. This feeling is confirmed looking at the references used in the introduction that come mostly from journals that deals with geological, marine and solid earth issues. Maybe the paper should be submitted to a journal that deal with these topics rather than HESS. I let the editorial board and the associated editor handle this question. Should this paper be published to HESS, I would like to see the following points be addressed in a revised version of the paper.

My first concern is related to the introduction of the paper. The introduction is too short and does not state properly the general context of the study and the research questions tackled. As a continental hydrologist, I wonder what fluid escapes and positive relief are used for. For what reasons should these systems be studied and identified? These critical points should be clearly explained in the introduction so that the reader can figure out the novelty/added-value of the research presented in the paper. It is very hard to understand if the identification of a positive relief and the explanation of its formation is a major research challenge or not. Moreover, it seems that no new methodology regarding theses question is proposed and that rather classical approaches were used. In this context, it is hard to state if the research presented is worth being published as it is.

The overall quality of the paper should also be improved as the description of the methods and the results are too short, lack precision and are sometime of poor quality. For instance, foraminiferal information is used to better understand the processes producing the positive relief. In the methodological part, it is not explained to what extent and how foraminifers can be used to better interpret these processes. Moreover, the text and figures are not always in agreement (see specific comments below). The conclusions drawn from the measurements of the isotopic composition of pore-water is not clear enough and should be discussed in greater details (see specific comment below). The same goes for the foraminiferal part.

I really like the fact that the authors have used a theoretical model to explain the processes responsible for the formation of the positive relief. I find this approach very valuable for the paper.

Specific comments:

- Line 41: "have been reported worldwide" - Line 82: "in the framework of a project entitled. . . sedimentological, geochemical and bathymetric studies . . . were performed". Please end the sentence with a point. - Please define what mbsl means here. - Line 135: I don't understand what the authors mean when they say that they measure pore water (w%). The unit is confusing. Even more confusing as the term water content is used when commenting Table 1 (line 162). - Line 164: TOC in the text and MOT in Table 1 and Figure 4. Please correct. - Line 174: this is not a sentence. . . - Line 193: pH is not shown Fig 5. - Comments on Figure 5 (line 186 to 189): There is indeed a trend to positive values but in my opinion, it is not as evident as the authors state. The authors also state that the delta 18O reaches a value of 6. This is more than questionable as (i) only the last point reaches this value and (ii) this point is clearly out of the trend. This point needs to be discussed thoroughly. - Line 237-239: Can this assertion be verified or confirmed by other measurements/approaches? - Line 261-271: These concluding remarks should be improved.

---

## Author Comment (AC1) · 19 Dec 2018

Reviewer comment: Introduction is too short Answer: We added new sentences, in which relationships between fluid escapes and positive relief are better explained in order to clarify the main challenge of our study.

Reviewer comment: Moreover, it seems that no new methodology regarding theses question is proposed and that rather classical approaches were used. In this context, it is hard to state if the research presented is worth being published as it is. Answer: The methodology novelty is the multi- and interdisciplinary approach to characterize completely the system, by using field and laboratory data, theory and modeling. We underline this approach adding a new sentence in Methods.

[Figure]

Reviewer comment: Methods and the results are too short Answer: In methods and results we include new sentences in order to improve the quality and understanding of the text

Reviewer comment: Foraminiferal analysis in methods are not explained Answer: In methods we include a sentence to explain the foraminiferal extraction and identification analysis.

Reviewer comment: The text and figures are not always in agreement (see specific comments below). Answer: The text and figures were edited by specific comments as suggested. Reviewer comment: The conclusions drawn from the measurements of the isotopic composition of pore-water is not clear enough and should be discussed in greater details (see specific comment below) Answer: We included more detailed and additional information to sections 2, 3 and 4 that support of our observations and discussion. Within this section, we now better highlight the analysis methodology (and acceptance criteria of the data), as well as, why we conclude that the results are not related to oceanic waters and most likely related to gas hydrate dissociation.

Reviewer comment: The authors also state that the delta 18O reaches a value of 6. Answer: We think that our formulation was not completely clear in the text, and that the reviewer misinterpreted our results. We stated that the maximum value for d18O was +1.8, the value close to 6 (5.6 exactly) refers to dD. We improve this sentence, to make this fact more explicit in the revised version.

Reviewer comment: This is more than questionable as (i) only the last point reaches this value Answer: This is true, this point is the most extreme one. However, we run the analysis of each sample at least twice, and in different days to avoid any analytical interference of the instrument. For each measurements, each sample is analyzed 5 consecutive times. For each run we accept a standard deviation less than 0.8 ‰ for Hydrogen and less than 0.1 ‰ for Oxygen. Than, the average of at least 2 different measurements should have the save standard deviation as before explained. In this

particular case we repeated 2 times. The first measurement was +5.9 (±0.26) ‰ while the second one was +5.32 (±0.56) ‰We included this information in the text, to make sure the reader also understand, how we performed the analysis and why we don't exclude this value from the results.

Reviewer comment: Line 261-271: These concluding remarks should be improved.check the quality of the measurements. (ii) this point is clearly out of the trend. This point needs to be discussed thoroughly. Answer: We consider that this point is indeed an extreme value, however as explained before a valid one. Moreover, if the value is excluded from the co-isotope correlation, the slope of the linear regression and its intercept only deviates slightly. With the point included the slope of the linear regression is 2.55 and the intercept is of +0.65, when this point is excluded the new slope is of 2.33 and the intercepts is +0.74. Therefore, we consider this point is not to be off-trend.

Reviewer comment: Line 261-271: These concluding remarks should be improved. Answer: We add new sentences in order to complete this part of conclusion by information regarding foraminifera and isotope results.

Please also note the supplement to this comment:
https://www.hydrol-earth-syst-sci-discuss.net/hess-2018-362/hess-2018-362-AC1-supplement.pdf

**Supplement:**

**Pore-water in marine sediments associated to gas hydrate dissociation offshore Lebu, Chile.**

Carolina Cárcamo[1,2], Iván Vargas-Cordero[1], Francisco Fernandoy[1], Umberta Tinivella[3], Alessandra Rivero[1], Diego López-Acevedo[4], Joaquim P. Bento[5], Lucía Villar-Muñoz[6], Nicole Foucher[1], Marion San Juan[1]

[1] Universidad Andres Bello, Facultad de Ingeniería, Quillota 980, Viña del Mar, Chile

[2] Centro de Investigación Marina Quintay. CIMARQ. Facultad de Ciencias de la Vida. Universidad Andres Bello, Viña del Mar, Chile.

[3] OGS Istituto Nazionale di Oceanografia e di Geofisica Sperimentale, Borgo Grotta Gigante 42/C, 34010, Sgonico, Italy.

[4] Universidad de Concepción, Departamento de Oceanografía, Programa COPAS Sur-Austral, Campus Concepción Víctor Lamas 1290, P.O. Box 160-C, Concepción, Chile

[5] Escuela de Ciencias del Mar, Pontificia Universidad Católica de Valparaíso, Av. Altamirano 1480, 2360007 Valparaíso, Chile.

[6] GEOMAR Helmholtz Centre for Ocean Research, Wischhofstr. 1-3, 24148 Kiel, Germany.

**ABSTRACT**

Gas hydrate occurrences along the Chilean margin has been widely documented, but the processes associated to fluid escapes originated by gas hydrate dissociation are as yet unknown. Here, we report a morphology growth related to fluid migration in the continental shelf offshore Lebu (37 °S) by analysing mainly geochemical features. In this study, oxygen and deuterium stable water isotopes in pore water were measured. The knowledge was completed by analysing bathymetric, biological and sedimentological data. From bathymetric interpretation, a positive relief at 127 m below sea level was recognized, oriented N55°E and characterised by five peaks. Moreover, enrichment values for $\delta^{18}O$ (from 0.0 to 1.8‰) and $\delta D$ (from 0.0 to 5.6‰) were obtained. These are typical values related to hydrate melting during coring and post-sampling. The evident orientation of the positive relief could be associated with faults and fractures already reported, which constitute pathways for fluid migration from deep to shallow zones. Finally, benthic foraminifera observed in the core sample can be associated to cold seep areas. Based on theoretical modelling, we conclude that the positive relief correspond to mud growing processes related to gas hydrates dissociation and represent a key area to investigate fluid migration processes.

Keywords: gas hydrate, stable isotopes, pore water, mud growing, fluid migration

**1. Introduction**

Morphological features associated with fluid escapes along continental margins (e.g. mud volcanoes, mud mounds, pockmarks and seeps) have been reported worldwide (e.g. Van Rensbergen et al., 2002; Loncke et al., 2004; Hovland et al., 2005; Lykousis et al., 2009; Chen et al., 2010). Fluid escapes can be formed mainly by biogenic and thermogenic methane gas and water. The gas can give place to gas hydrate formation in marine sediments if pressure and temperature conditions are adequate (Sloan, 1998), in which the gas is trapped in a lattice of water molecules. Gas hydrate occurrences along the Chilean margin are distributed from 33 to 57°S (Bangs et al., 1993; Froelich et al., 1995; Morales, 2003; Grevemeyer et al., 2003; Rodrigo et al., 2009; Vargas-Cordero et al., 2010, 2010a, 2016, 2017; Villar-Muñoz et al., 2014, 2018, 2018a). Several studies have documented fluid escapes related to gas hydrate dissociation through faults and fractures (e.g., Yin et al., 2003; Thatcher et al., 2013).

Identification of areas where gas hydrate dissociation processes is occurring play an important role because allow us to map shallow fluid escapes zones, in which the methane, known as a potent greenhouse gas (IPCC, 2014), can contribute to: a) increase the temperature and take part in the global warming; b) change the physico-chemical conditions of the seawater; c) affect the marine microfaunal diversity pattern; and d) the nucleation and rupture propagation of earthquakes (Sibson, 1973; Rathburn et al., 2003; Thatcher et al., 2013; Ruppel and Kessler, 2017). Among others, common techniques often used to recognize such processes are: biological, geochemical and geophysical analyses. Biological indicators as benthic foraminifera, bivalve shells and microbial communities have been related with fluid escapes (Reed et al., 2002; Chen et al., 2007; Karstens et al., 2018). For example, foraminiferal taxa reported worldwide that include Uvigerina, Bolivina, Chilostomella, Globobulimina, and Quinqueloculina can be related with cold seep occurrences where a high food availability attract them (Bernhard et al., 2000; Rathburn et al., 2000; Hill et al., 2003). Moreover, enriched stable water isotope values have been measured from pore water in marine sediments. Hesse (2003) and Kvenvolden and Kastner (1990) reported in extensive articles several cases of enriched stable water isotope values from different regions, including the Chilean coast. Finally, geophysical studies have allowed identifying morphologies associated with fluid escapes by using bathymetric, backscatter and high resolution images (Sager et al., 2003; Loncke et al., 2004; Tinivella et al., 2007). Well and seismic data interpretations allowed recognizing an active structural domain offshore Arauco basin (Melnick and Echtler, 2006; Melnick, 2006a). During the depositional history of Arauco Basin, numerous tectonic phases have been recognized, including subsidence and uplift episodes that gave place to accretion and erosion of the prism (Bangs and Cande, 1997; Lohrmann, 2002). Cretaceous-Plio-Pleistocene marine and continental sequences configure a cyclic sedimentary complex. Sedimentary sequences are composed by alternating of marine and continental deposits. From base to top, these are: Quiriquina (Biró-Bagóczky, 1982), Pilpilco, Curanilahue, Boca Lebu, Trihueco, Millongue, Ranquil, Tubul and Arauco formations (Pineda, 1983; Viyetes et al., 1993; Muñoz-Cristi, 1956; Muñoz-Cristi, 1968). The Nahuelbuta Range is composed by Carboniferous-Permian granitoides (Coastal Batholith) intruding the Paleozoic-Mesozoic metamorphic rocks. Moreover, gas and carbon reservoirs have been identified along the Arauco basin (Mordojovich, 1974; González, 1989).

This study aims at characterizing a positive relief identified in order to understand its origin by using geochemical, sedimentological and bathymetric data. The study area is located in the continental shelf, ~150 meters below sea level (mbsl) and includes part of the Arauco basin (Fig. 1).

**2. Data and Methods**

**2.1 Data**

In the framework of the project entitled "Identification and quantification of gas emanations associated with gas hydrates (FONDECYT 11140214)" sedimentological, geochemical and bathymetric studies offshore Lebu were performed (Fig. 1). In 2016 and 2017 two marine campaigns onboard R/V Kay Kay II were carried out collecting bathymetric data, seawater samples and marine sediments.

Marine sediment samples were recovered by using a gravity corer (diameter equal to 9 cm) at around 127 mbsl and it drilled as deep as 240 cm into marine sediments (core GC-02). The core collected is located around a positive relief close to 73°44'25"W-37°36'10"S (Fig. 2). The GC-02 core was divided into four sections of 60 cm long (S01, S02, S03, S04); each section was frozen and analysed at the Sedimentology Andres Bello University's laboratory (Viña del Mar, Chile).

The water samples were collected by Niskin bottles at five depths (0 m, 10 m, 20 m, 50 m and seafloor); temperature, conductivity, dissolved oxygen and pH was determined with the multiparameter measurer model IP67. These parameters were measured at the two ends of the identified lineament, i.e., the first station located to the south and the second one to the north (Fig. 2).

**2.2 Methods**

The procedure is based in a multi- and interdisciplinary approach to completely characterize the system, by using field and laboratory data, theory and modelling. The approach includes: a) bathymetric data processing and b) sedimentological, physical-chemical, geochemical and biological analyses of seawater and marine sediment samples.

Bathymetric and sound velocity data were acquired using a multihaz Reson SeaBat 7125 echosounder (400 kHz, 0.5° x 1°), a SVP90 probe, and an AML Oceanographic Model Minos X sound velocity profiler. A preliminary processing was performed onboard using a PDS2000 commercial software, which allows correcting bathymetric data in real time using the SVP90, AML information and ship motions (pitch, roll, yaw and heave). The bathymetric data processing was performed using the open-source MB-System software (Caress and Chayes, 2017). In this step, bathymetric data were converted in MB-System format in order to attenuate tide and scattering effects. In the first step, bathymetric grids were created with nearneighbor interpolation algorithm, using the open-source software Generic Mapping Tools (GMT, Wessel et al., 2013). The algorithm builds cell values in depth rectangular distributed, in which each node value corresponds to the weighted average of around probes

of search circle of 1 arc second. Besides the selected grid was configured with spatial resolution of 0.2 arc seconds. Finally, a median filter of 10 m width was applied in order to smooth the grid.

Grain size analysis includes sieving method where sediments pass (by agitation) through meshes; in our case, 50 g of sediment samples were sieved by using the following mesh sizes: 60, 80, 120 and 230. The pipette method was adopted in order to separate clay and silt fractions by selecting 15 g of mud sample. Statistical parameters were calculated in agreement with reported formulas (Folk and Ward, 1957; Carver, 1971; Scasso and Limarino, 1997).

Seawater physical-chemical properties (temperature, pH, salinity and dissolved oxygen) in proximity of the positive relief were obtained using the multiparameter Meter (IP67, model 8602). The multiparameter Meter has different types of probes or electrodes, which must be selected according to the required function to obtain accurate measurements. The temperature was measured in Celsius degree, with an accuracy of ±0.5°C, while pH was directly related to the ratio of the concentrations of hydrogen ions [$H^+$] and hydroxyl [OH] (Cabo, 1978) with an accuracy of ±0.1. Salinity was obtained from the conductivity, which depends on the number of dissolved ions per unit volume and the mobility of the ions; the accuracy is ±0.1. Finally, dissolved oxygen can be measured both in % and in mg/L, with an accuracy of ±3%; in our case, it was expressed in %.

The core was cut in sections of 10 cm long and then the main physical-chemical parameters were measured including water content (%), porosity ($\Phi$), content of solid material per unit volume, expressed as apparent density ($\rho$; Salamanca and Jara, 2003) and total organic carbon (TOC). Finally, the samples were dried in forced air oven at 60°C for 36 hours and in a desiccator for 30 minutes.

TOC content was measured by gravimetric determination of weight loss through loss-on-ignition method (Byers et al., 1978; Luczak et al., 1997). In our case, 2 g of dry sediment sample was calcined in a muffle at 500 °C for 5 hours and then it was placed in desiccator for 30 minutes until to register constant weight in order to reduce the associated error.

For the foraminifera extraction, the corer was cutted into sections of 15 cm from which 50 g of material was extracted, which was washed, dried and sieved, 120 and 230 sieves were used. The specimens were extracted under binocular magnification, being deposited in Petri dishes separated by group for their later identification. The general morphological features of the specimens were classified using the Atlas of Benthic Foraminifera (Hobourn et al., 2013) and the genus was identified based on the study of Chilean material (e.g. Figueroa et al., 2005).

Pore water from core was extracted using an ACME lysimeter (0.2 μm) in order to analyse oxygen and deuterium stable water isotopes. The pore water extraction procedure includes: a) corer cutting in sections of 5 cm long; b) centrifugation; c) pore water extraction by using Rhizon MOM with pore sizes ranging from 0.12 to 0.18 μm; and d) stable water isotope

determination by Cavity Ring Down Spectroscopy (CRDS) method at the Laboratorio de Análisis Isotópico (LAI) at the Universidad Andrés Bello (Viña del Mar, Chile).

Oxygen and deuterium water isotope analyses were evaluated using LIMS (Coplen and Wassenaar, 2015) and normalized to the VSMOW-SLAP scale and reported as δ-values for oxygen ($\delta^{18}O$) and deuterium ($\delta D$). Each sample was measured at least twice in different days. For each measurement, samples were analysed for five consecutive times. Results are accepted if the standard deviation of each single run (composed of five repetitions) is <1‰ for $\delta D$ and <0.1‰ for $\delta^{18}O$. Thereafter, the accepted stable water isotope value of a sample, will be the average of the (at least) two different valid measurements within the range of the previously explained standard deviation.

**3. Results**

From bathymetric data, a positive relief located at 127 mbsl with orientation N55°E was recognised. The relief shows an elevation of about 6 m above the seafloor, an extension of 410 m length and a width of 50 m reaching an area of 14640 $m^2$ (Fig. 2). Five peaks ranging from 3 to 9 m high along the relief were observed.

Grain size analysis shows constant values with depth. The value of the average grain size corresponds to sandy mud textural group. Silt size reaches 60% of total volume (Fig. 3). Physical-chemical parameter distributions of core GC-02 are detailed in Table 1. A slightly variation of water content (W) ranging from 38.6 to 46.3 % (average equal to 43.1%), porosity (ϕ) ranging from 62.7 to 69.7 % (average equal to 66.9%) and apparent density (ρ) ranging from 1.5 to 1.7 g/$cm^3$ (average equal to 1.6 g/$cm^3$) were detected. TOC values show a variable trend with maximum value equal to 8.7% of total volume located at 2.2 m, while the minimum value is equal to 5.1% of total volume detected at 0.4 m (Fig. 4). Note, as expected, an opposite trend distribution was recognized between porosity and apparent density.

Pore stable water isotope analysis of the marine sediment core shows positive values ranging from 0.0 up to +1.8‰ for of $\delta^{18}O$ and 5.6‰ for $\delta D$, respectively (Fig 5). Stable water isotope δ-values show a positive trend (enrichment) towards the bottom of the sediment core, with values close to 0 at the top in the sediment-seawater interface, and a restricted variability for all samples analysed (Std. Dev. 0.33 and 0.95 for $\delta^{18}O$ and $\delta D$, respectively). It was noticed that no negative values were found along the core.

Benthic foraminiferal accumulations were found in shallower levels of the core (0-60 cm) showing globose and elongated morphologies. The following genus of opportunistic foraminifera were recognized: Globobulimina, Bolivina, Anomalinoides, Uvigerina, Oridorsalis and Quinqueloculina (Fig. 6).

Respect to the properties of the water column, temperature values range from 12 to 14 °C in seawater samples, registering maximum values in correspondence of shallower levels, while minimum values were found in deeper levels. Salinity and dissolved oxygen values

show a similar trend with maximum values equal to 33‰ and 60% located at 20 mbsl, respectively. Minimum values of salinity (31 ‰) and dissolved oxygen (66.2%) were measured in station 1 (see Fig. 2 for location) at 0.6 mbsl. pH values range from 7.5 to 8.1.

**4. Discussion and conclusion**

The stable water isotope composition of pore water represents a strong evidence of gas-hydrate dissociation. Figure 5a shows the stable water isotope profile of the entire core, showing a clear increase with depth, with values close to 0‰ at seawater-sediment interface to positive values at the bottom of the core (1.8‰ for $\delta^{18}O$ and 5.6‰ for $\delta D$). According to observational data for similar latitudes and modelled surface water stable isotope composition for this ocean region, shallow water should have a slightly negative isotope composition (~-0.2 to -0.5‰ $\delta^{18}O$) (Schmidt et al., 1999; LeGrande and Schmidt, 2006), which are related to the transport of Subantarctic Waters through the Humboldt Current System along the Chilean coast (Silva et al., 2009). The given negative values are mainly influenced by the mix of oceanic and depleted melt water from the Antarctic Ice Sheets (Sharp, 2007). Therefore, the observed trend shows the influence of seawater mixing on the top and a different source at the bottom of the core. Positive values of meteoric waters are mostly associated to high evaporation rates, which must be discarded in the context of this investigation. Positive $\delta^{18}O$ values have been reported for clay minerals dewatering; however, in this case a $\delta D$ depletion rather than enrichment is expected (Hesse, 2003). Nonetheless, the co-isotope relationship (Fig. 5b) of our samples shows that pore waters stable water composition have a strong ($r^2$=0.8) positive correlation (i.e. simultaneous enrichment of $\delta^{18}O$ and $\delta D$). Additionally, the meteoric origin of the pore water can be rejected as shown in Fig. 5c, as pore waters fall away from the Global Meteoric Water Line (GMWL), which defines the fractionation processes during the hydrological cycle (Craig, 1961). Stable water isotope enrichment of pore has been related to hydrate melting during coring and post-sampling (Hesse, 2003; Tomaru et al., 2006), which are preferentially enriched by heavy stable water isotope.

The infaunal foraminifera, found in the shallower sediment samples (e.g. Bolivina Globobulimina and Uvigerina), could be associated with modern cold seeps, since they can metabolize seeping methane, directly or indirectly exploiting the available geochemical energy source (Jones, 2014). In fact, several authors reported these genus over the continental margins of California, Japan and Mexico (Bernhard et al., 2000; Rathburn et al., 2000; Hill et al., 2003), where these benthic communities were identified living over seepage areas. Besides, benthic foraminifera are associated with high organic content ambient, low oxygen conditions and cold seep occurrences (Hill et al, 2003; Rathburn et al. 2000).

[revised manuscript text omitted]

IPCC.: Contribution of Working Groups I, II and III to the Fifth Assessment Report of the Intergovernmental Panel on Climate Change. In Climate Change 2014: Synthesis Report; Pachauri, R.K., Meyer, L.A., Eds.; IPCC: Geneva, Switzerland, pp. 1–151. ISBN 978-92-9169-143-2, 2014.

[revised manuscript text omitted]

**Figure 7**: Schematic profile explaining mud growing formation (in red). The profile location is shown in Fig. 1. Dashed lines show theoretical bases of GHSZ by using geothermal gradient of 30°C/km for several scenarios supposing that the hydrate is formed by a mixture of 95% of methane and 5% of ethane. The blue dashed line indicates the actual theoretical base of the GHSZ. The dotted lines indicate the theoretical base of GHSZ supposing a decrease of the bottom temperature of 2 °C (black dotted line), 3 °C (magenta dotted line), 4 °C (green dotted line) and 5 °C (red dotted line). The black solid line indicates the seafloor. The pink arrows indicate the direction of the fluid/mud outflow. Possible faults and fractures are also reported as black lines.

| Depth (m) | W (%) | φ (%) | ρ (g/cm3) | TOC % |
|---|---|---|---|---|
| 0,1 | 45,2 | 68,8 | 1,56 | 7,9 |
| 0,2 | 42,2 | 66,1 | 1,61 | 7,2 |
| 0,3 | 41,2 | 65,2 | 1,62 | 6,6 |
| 0,4 | 41,3 | 65,3 | 1,62 | 5,1 |
| 0,5 | 39,9 | 64,0 | 1,64 | 5,9 |
| 0,6 | 38,6 | 62,7 | 1,67 | 6,0 |
| 0,7 | 40,3 | 64,4 | 1,64 | 5,9 |
| 0,8 | 42,7 | 66,5 | 1,60 | 6,4 |
| 0,9 | 43,0 | 66,8 | 1,60 | 6,2 |
| 1 | 42,8 | 66,6 | 1,60 | 6,5 |
| 1,1 | 42,6 | 66,5 | 1,60 | 6,6 |
| 1,2 | 42,9 | 66,7 | 1,60 | 6,1 |
| 1,3 | 45,0 | 68,6 | 1,57 | 6,0 |
| 1,4 | 45,2 | 68,8 | 1,56 | 6,7 |
| 1,5 | 45,1 | 68,7 | 1,56 | 4,2 |
| 1,6 | 46,3 | 69,7 | 1,55 | 7,5 |
| 1,7 | 44,1 | 67,8 | 1,58 | 6,7 |
| 1,8 | 45,3 | 68,8 | 1,56 | 7,1 |
| 1,9 | 40,7 | 64,7 | 1,63 | 5,4 |
| 2 | 43,7 | 67,5 | 1,58 | 7,0 |
| 2,1 | 44,3 | 68,0 | 1,58 | 6,8 |
| 2,2 | 43,3 | 67,1 | 1,59 | 8,7 |
| 2,3 | 45,4 | 68,9 | 1,56 | 6,9 |
| 2,4 | 44,4 | 68,0 | 1,58 | 6,5 |
| Average | 43,1 | 66,9 | 1,59 | 6,5 |
| Minimum | 38,6 | 62,7 | 1,55 | 4,2 |
| Maximun | 46,3 | 69,7 | 1,67 | 8,7 |

Table 1: Physical-chemical parameter distribution in marine sediments

---

## Referee Comment (RC2) · Anonymous Referee #2 · 18 Jan 2019

The manuscript on 'Pore-water in marine sediments associated to gas hydrate dissociation offshore Lebu, Chile, submitted by Carcamo et al., presents a sequence of observations/data and results relying on a multiproxy approach/investigation of a sediment core collected in a site located in the vicinity of a relief disposed along the Chilean margin. From the interpretation of bathymetric, sediment, geochemical and foraminifera data, in combination with a theoretical model, the authors intend to prove that the positive relief corresponds to mud growing related to gas hydrate dissociation.

General comments

While this manuscript intends to provide evidence supporting the assumption that the relief along the Chilean margin (as shown in Figure 2) is related to gas hydrate dissociation, the data presented essentially fails to do so. Data and information on pore water,

sediment and foraminifera do not support the assumption that gas hydrate or methane emissions may actually explain the formation of such a relief. The main conclusions therefore seem to be rather far-fetched – at least in the current form of the manuscript. Several aspects, such as the use of forminifera to explain a relief, the integration of water samples data with sedimentological data, need further clarification and proper interpretation. Other open questions relate to the location of the sulphate-methane-transition zone (SMTZ); when was the relief actually formed (if really related to gas hydrate – note that the sediment core used to conclude on the presence of gas hydrate appears to be located (too) far off the relief to be really considered as a relevant and reliable indicator); are there any indications of a gas hydrate reservoir in the seismic data? Some of the aspects and open questions might benefit from further information in the methodology section of the manuscript. For example, there is a lack of information on the micropaleontological methods used and the exact origin of the results/data used in this study. For stable isotope measurements no errors are given, nor the standards used. There is also needed further information on the theoretical model used in this study. The references used in the discussion chapter should be updated with more recent and relevant publications. In its current form, this contribution has major shortcomings that would have to be addressed before any further re-consideration for publication in this journal. The topic, data presented, results and conclusions furthermore are actually quite far from the actual main scope of the HESS journal. The authors would have to do an additional substantial effort to have their contribution really target the audience of HESS. On the technical side, the manuscript also suffers from quite substantial shortcomings. There is quite a large amount of typing errors and poor English language that need to be improved. The quality of some figures (especially Figure 6) also needs to be improved.

Specific comments:

Line 44: when referring to biogenic and thermogenic methane gas, please note that thermogenic is also biogenic, as it stems from organic matter degradation. An alterna-

tive would be to use the term 'microbial' rather than 'biogenic'.

Line 45: Conditions proper for gas hydrate formation need to be further explained in this part of the manuscript.

Lines 54-56: Note that the organisms listed here cannot be considered as biological indicators of fluid escape per se.

Lines 56-59: Please specify in which stable isotopes the pore water has been enriched.

Line 89: "located around a positive relief" is not clear. Please be more specific.

Lines 93-99: Please develop on how the water samples collected in the water column are related to sediment features?

Line 175: Foraminifera do not have gender (genus instead?).

Line 185-186: The data shown here does not support gas hydrate dissociation. The analysed core is located too far from the relief.

Line 188. In Fig. 5a there are not values up to 6‰

Lines 194-202. In this section statements are rather confusing and need to be reformulated.

Lines 203-208: Since methane is not metabolized by foraminifera, they cannot be used for discriminating between seeping and non-seeping sites.

Line 220: geothermal gradient to be expressed in 0C/km not km/0C.

Lines 240-247: From the current developments, it is not clear how grain size data fits into this paper and how it can serve sustaining the main hypothesis.

Lines 248-260: Similar comment than here above, but for seawater data.

Lines 261-264: It is not clear how the data used in this paper supports the main hypothesis (i.e. the observed relief is related to 'fluid flux channelized by faults and fractures'.

Nor does the data provide evidence for gas hydrate dissociation.

Line 537: The term 'co-isotope distribution' is confusing. Rather use 'stable isotopic composition' instead.

Fig. 6. The pictures are rather blurry. Moreover, those are common species that should to be classified. The name of the genus is not enough.

Fig. 7. Seismic data would be needed to really assess this schematic profile.

---

## Author Comment (AC2) · 7 Feb 2019

Answers Reviewer 2

1) While this manuscript intends to provide evidence supporting the assumption that the relief along the Chilean margin (as shown in Figure 2) is related to gas hydrate dissociation, the data presented essentially fails to do so. Data and information on pore water, sediment and foraminifera do not support the assumption that gas hydrate or methane emissions may actually explain the formation of such a relief. The main conclusions, therefore, seem to be rather far-fetched – at least in the current form of the manuscript.

Thank you for your comment. We had better support our hypothesis providing more details and better underlying that different pieces of evidence like geochemical, biological and geophysical arguments point out to gas hydrate dissociation. Our hypothesis about mud growing related to gas hydrate dissociation is strongly supported by: a) The benthic foraminifera presence that has proved to be a good indicator of methane releases (Sen Gupta and Aharon., 1994; Rathburn et al., 2000; Hill et al., 2003; Torres et al., 2003) b) The enrichment isotopic composition of O18 and D. c) The theoretical modelling. d) The gas hydrate presence in the study area, as indicated by the BSR presence reported in Vargas-Cordero et al., (2010a, 2011).

2) Several aspects, such as the use of foraminifera to explain a relief);

We include new sentences related to Foraminifera in the Introduction and discussion in order to explain its relation with our hypothesis.

3) the integration of water samples data with sedimentological data, need further clarification and proper interpretation.

Regarding the water and sedimentological samples were analysed in order to evaluate the gas hydrate presence. In fact, sediments in presence of gas hydrate affect the physical-chemical properties at the interface seawater-subbottom. We improve our Methods and discussion including more detail.

4) Other open questions relate to the location of the sulphate-methane-transition zone (SMTZ);

Thank you for rising this point. We are aware that it might be possible to associate the process occurring inside the SMTZ with mud growing, but our measurements and analyses do not support us to confirm this. It means, we did not measure sulfate, methane, sulfate reduction or AOM (anaerobic oxidation of methane). After saying that, our hypothesis is strongly supported by BSR (Fig. 7), isotopic composition (O18 and D), foraminifera taxa, and static modelling. To highlight the gas hydrate presence in the study area, we added a new figure (Fig. 7) where the BSR was identified.

5) when was the relief actually formed (if really related to gas hydrate)

Radiometric dating analyses were not part of our goals; however, our hypothesis is strongly supported as previously explained.

6) note that the sediment core used to conclude on the presence of gas hydrate appears to be located (too) far off the relief to be really considered as a relevant and reliable indicator

Our sediment core is located near (100 m) to relief, also the study reported by Rivero (2018) shows sediment cores on the relief with similar results. We include new sentences in the Discussion in which the data of Rivero is referenced.

7) are there any indications of a gas hydrate reservoir in the seismic data?

We included a new Figure (Fig. 7), in which we reported the bottom simulating reflector (BSR) as identified in a seismic line (SO161-44) analysed by Vargas-Cordero et al., (2010a, 2011). In fact, the BSR indicates the base of the gas hydrate zone and the top of the below free gas zone. So, Fig. 8 underlines the presence of gas hydrate in the study area, as already detected and analysed; consequently, the Discussion was coherently modified and updated.

8) Some of the aspects and open questions might benefit from further information in the methodology section of the manuscript. For example, there is a lack of information on the micropaleontological methods used and the exact origin of the results/data used in this study.

Thank you for your comment. We included in Methods and Discussion information about foraminifera analysis.

9) For stable isotope measurements, no errors are given, nor the standards used.

Additional information was added to the reviewed version. Nonetheless to clarify this point, We will add the following information "Oxygen and deuterium water isotope analyses were evaluated using LIMS (Coplen and Wassenaar, 2015) and normalized to the VSMOW-SLAP scale and reported as $\delta$-values for oxygen ($\delta$18O) and deuterium ($\delta$D). Each sample was measured at least twice in different days. For each measurement, samples were analysed for five consecutive times. Results are accepted if the standard deviation of every single run (composed of five repetitions) is <1‰ for $\delta$D and <0.1‰ for $\delta$18O. Thereafter, the accepted stable water isotope value of a sample will be the average of the (at least) two different valid measurements within the range of the previously explained standard deviation."

10) There is also needed further information on the theoretical model used in this study.

We included details about the method used to evaluate the theoretical BSR depth, as required.

11) The references used in the discussion chapter should be updated with more recent and relevant publications.

We included new references and the reference list has been updated as required.

12) In its current form, this contribution has major shortcomings that would have to be addressed before any further re-consideration for publication in this journal. The topic, data presented, results and conclusions furthermore are actually quite far from the actual main scope of the HESS journal. The authors would have to do an additional substantial effort to have their contribution really target the audience of HESS.

We improved our article, modifying Introduction, Methods and Discussions in order to make our manuscript more attractive to the HESS audience.

13) On the technical side, the manuscript also suffers from quite substantial shortcomings. There is quite a large amount of typing errors and poor English language that need to be improved. The quality of some figures (especially Figure 6) also needs to be improved.

We improved the text correcting shortcomings and the figures as suggested.

14) Specific comments:

Line 44: when referring to biogenic and thermogenic methane gas, please note that thermogenic is also biogenic, as it stems from organic matter degradation. An alternative would be to use the term 'microbial' rather than 'biogenic'. We edited as suggested.

Line 45: Conditions proper for gas hydrate formation need to be further explained in this part of the manuscript. We added a new sentence in Introduction as suggested.

Lines 54-56: Note that the organisms listed here cannot be considered as biological indicators of fluid escape per se. We included more information regarding foraminifera taxa as an indicator of methane release.

Lines 56-59: Please specify in which stable isotopes the pore water has been enriched. We included more details as well as the values of O18 and D.

Line 89: "located around a positive relief" is not clear. Please be more specific. We modified the sentence by giving more details.

Lines 93-99: Please develop on how the water samples collected in the water column are related to sediment features? We improved our discussion including information about relationships between the water column and sediments. Also, we included a new sentence in Methods in order to clarify relationships between seawater and sea bottom.

Line 175: Foraminifera does not have gender (genus instead?). We changed as suggested.

Line 185-186: The data shown here do not support gas hydrate dissociation. The analysed core is located too far from the relief. Our sediment cores are located near (100 m) the relief. We included details about a recent study reported by Rivero (2018) that reports sediment cores on the relief with similar results. We included new sentences in the Discussion in order to underline this point.
Line 188. In Fig. 5a there are not values up to 6‰Ẇe specified the values in the Results and Discussion.

Lines 194-202. In this section, statements are rather confusing and need to be refor-mulated. We completely modified the text as suggested.

Lines 203-208: Since methane is not metabolized by foraminifera, they cannot be used for discriminating between seeping and non-seeping sites. We specified the foraminifera taxa reported as an indicator of gas release.

Line 220: geothermal gradient to be expressed in 0C/km not km/0C. We corrected the mistake.

Lines 240-247: From the current developments, it is not clear how grain size data fits into this paper and how it can serve to sustain the main hypothesis. The grain size is related to the geological context that promote the hydrate formation. Gas hydrate is believed to exist in various forms within muddy layers or embedded within the pores of sandy layers (i.e., Waite et al.,2009; Hyodo et al., 2013). A comprehensive database of measurements for hydrate-bearing clay, silt, and sand at different effective stress and hydrate saturation levels is documented by Lee (2007). Moreover, the grain size can be associated with flow hydrodynamic conditions, in which mud and sand could be related to coastal and beach systems, fluvial or deltaic deposits, confirmed by the TOC value. So, the presence of the gas hydrate is compatible with the geological context.

Lines 248-260: Similar comment than here above, but for seawater data. The seawater data were analysed in order to evaluate the salinity and temperature at the interface seawater and sub-bottom. For example, in the case of freshwater spring from gas hydrate dissociation, the salinity would decrease.

Lines 261-264: It is not clear how the data used in this paper supports the main hypoth-esis (i.e. the observed relief is related to 'fluid flux channelized by faults and fractures'. Nor do the data provide evidence for gas hydrate dissociation. We better explain that all data available (including data in literature) suggests that fractures and faults should be present and related to the fluid flux that could generate the relief.

Line 537: The term 'co-isotope distribution' is confusing. Rather use 'stable isotopic composition' instead. In figure 5b and 5c we refer to the linear relationship between $\delta$D and $\delta$18O, and not only to the stable water isotope composition. It's important to remark characteristics like slope and intercept of the line. We changed this to: "b. co-isotope linear regression of pore water samples and c. Co-isotope relationship of pore water samples against the global meteoric water (GMWL)".

Fig. 6. The pictures are rather blurry. Moreover, those are common species that should to be classified. The name of the genus is not enough. The pictures were improved and it was possible to identify some specimens at the species level, according to the bibliography of the study area.

Fig. 7. Seismic data would be needed to really assess this schematic profile. We included information about the BSR reported in this zone adding new figure (Fig. 7) modified from Vargas-Cordero et al. (2011).

Please also note the supplement to this comment:
https://www.hydrol-earth-syst-sci-discuss.net/hess-2018-362/hess-2018-362-AC2-supplement.pdf

―――――――――――――――

**Supplement:**

**Pore-water in marine sediments associated with gas hydrate dissociation**

**offshore Lebu, Chile.**

Carolina Cárcamo[1,2], Iván Vargas-Cordero[1], Francisco Fernandoy[1], Umberta Tinivella[3],
Alessandra Rivero[1], Diego López-Acevedo[4], Joaquim P. Bento[5], Lucía Villar-Muñoz[6], Nicole
Foucher[1], Marion San Juan[1]

[1] Universidad Andres Bello, Facultad de Ingeniería, Quillota 980, Viña del Mar, Chile

[2] Centro de Investigación Marina Quintay. CIMARQ. Facultad de Ciencias de la Vida.
Universidad Andres Bello, Viña del Mar, Chile.

[3] OGS Istituto Nazionale di Oceanografia e di Geofisica Sperimentale, Borgo Grotta Gigante
42/C, 34010, Sgonico, Italy.

[4] Universidad de Concepción, Departamento de Oceanografía, Programa COPAS Sur-
Austral, Campus Concepción Víctor Lamas 1290, P.O. Box 160-C, Concepción, Chile

[5] Escuela de Ciencias del Mar, Pontificia Universidad Católica de Valparaíso, Av. Altamirano
1480, 2360007 Valparaíso, Chile.

[6] GEOMAR Helmholtz Centre for Ocean Research, Wischhofstr. 1-3, 24148 Kiel, Germany.

**19 ABSTRACT**

Gas hydrate occurrences along the Chilean margin has been widely documented,
but the processes associated with fluid escapes originated by gas hydrate
dissociation are yet unknown. Here, we report a morphology growth related to fluid
migration in the continental shelf offshore Lebu (37 °S) by analysing mainly
geochemical features. In this study, oxygen and deuterium stable water isotopes in
pore water were measured. The knowledge was completed by analysing
bathymetric, biological and sedimentological data. From bathymetric interpretation,
a positive relief at 127 m below sea level was recognized, oriented N55°E and
characterised by five peaks. Moreover, enrichment values for $\delta^{18}O$ (from 0.0 to
1.8‰) and $\delta D$ (from 0.0 to 5.6‰) were obtained. These are typical values related to
hydrate melting during coring and post-sampling. The evident orientation of the
positive relief could be associated with faults and fractures already reported, which
constitute pathways for fluid migration from deep to shallow zones. Finally, benthic
foraminifera observed in the core sample can be associated with cold seep areas.
Based on theoretical modelling, we conclude that the positive relief corresponds to
mud growing processes related to gas hydrates dissociation and represents a key
area to investigate fluid migration processes.

**Keywords**: gas hydrate, stable isotopes, pore water, mud growing, fluid migration

**1. Introduction**

Morphological features associated with fluid escapes along continental margins (e.g. mud volcanoes, mud mounds, pockmarks and seeps) have been reported worldwide (e.g. Van Rensbergen et al., 2002; Loncke et al., 2004; Hovland et al., 2005; Lykousis et al., 2009; Chen et al., 2010). Fluid escapes can be formed mainly by microbial and thermogenic methane gas and water. The gas can give place to gas hydrate formation in marine sediments if pressure and temperature conditions are adequate (Sloan, 1998), in which the gas is trapped in a lattice of water molecules. Along the continental margins gas hydrates occur naturally in gas hydrate stability zone (GHSZ), at ocean water depths greater than 300–500 m with low temperature, high pressure and adequate amounts of sedimentary organic carbon (2 to 3.5%), wherever enough methane is present. Moreover, the salinity, gas composition, geological structure, fluid migration and pore space of marine sediments are factors influencing gas hydrate formation (Ginsburg and Soloviev, 1998; Fehn et al., 2000; Dickens, 2001). Gas hydrate occurrences along the Chilean margin are distributed from 33 to 57°S (Bangs et al., 1993; Froelich et al., 1995; Morales, 2003; Grevemeyer et al., 2003; Rodrigo et al., 2009; Vargas-Cordero et al., 2010, 2010a, 2016, 2017; Villar-Muñoz et al., 2014, 2018, 2019). Several studies have documented fluid escapes related to gas hydrate dissociation through faults and fractures (e.g., Yin et al., 2003; Thatcher et al., 2013).

Identification of areas where gas hydrate dissociation processes are occurring play an important role because allow us to map shallow fluid escapes zones, in which the methane, known as a potent greenhouse gas (IPCC, 2014), can contribute to: a) increase the temperature and take part in the global warming; b) change the physico-chemical conditions of the seawater; c) affect the marine microfaunal diversity pattern; and d) affect the nucleation and rupture propagation of earthquakes (Sibson, 1973; Rathburn et al., 2003; Thatcher et al., 2013; Ruppel and Kessler, 2017). Among others, common techniques often used to recognize such processes are: biological, geochemical and geophysical analyses. Biological indicators as benthic foraminifera, bivalve shells and microbial communities have been related with fluid escapes (Sen Gupta and Aharon, 1994; Torres et al., 2003; Reed et al., 2002; Chen et al., 2007; Karstens et al., 2018). For example, foraminiferal taxa reported worldwide that include Uvigerina, Bolivina, Chilostomella, Globobulimina, Quinqueloculina, and Nonionella can be related with cold seep occurrences and methane presence, which can live in such conditions of organic-rich and reducing environment where a high food availability attract them (Bernhard et al., 2000; Rathburn et al., 2000; Hill et al., 2003). Moreover, enriched stable water isotope values have been measured from pore water in marine sediments. Tomaru et al., (2006), Hesse (2003) and Kvenvolden and Kastner (1990) reported in extensive articles several cases of enriched stable water isotope values from different regions related to gas hydrate dissociation, including the Chilean coast. The reported values for $\delta^{18}O$ and $\delta D$ reaches up to 3‰ and 10‰, respectively. Finally, geophysical studies have allowed identifying morphologies associated with fluid escapes by using bathymetric, backscatter and high-resolution images (Sager et al., 2003; Loncke et al., 2004; Tinivella et al., 2007). Besides, well and seismic data interpretations allowed to identify an active structural domain offshore Arauco basin (Melnick and Echtler, 2006; Melnick, 2006a). During the depositional history of Arauco Basin, numerous tectonic phases have been recognized, including subsidence and uplift episodes that gave place to accretion and erosion of the prism (Bangs and Cande, 1997; Lohrmann, 2002). Cretaceous-Plio-Pleistocene marine and continental sequences configure a cyclic sedimentary complex. The sedimentary sequences are composed by alternating of marine and continental deposits. From base to top, these are: Quiriquina (Biró-Bagóczky, 1982), Pilpilco, Curanilahue, Boca Lebu, Trihueco, Millongue, Ranquil, Tubul and Arauco formations (Pineda, 1983; Viyetes et al., 1993; Muñoz-Cristi, 1956; Muñoz-Cristi, 1968). The Nahuelbuta Range is composed by Carboniferous-Permian granitoid (Coastal Batholith) intruding the Paleozoic-Mesozoic metamorphic rocks. Moreover, gas and carbon reservoirs have been identified along the Arauco basin (Mordojovich, 1974; González, 1989).

This study aims at characterizing a positive relief identified in order to understand its origin by using geochemical, sedimentological and bathymetric data. The study area is located on the continental shelf, ~150 meters below sea level (mbsl) and includes part of the Arauco basin (Fig. 1).

**2. Data and Methods**

**2.1 Data**

In the framework of the project entitled "Identification and quantification of gas emanations associated with gas hydrates (FONDECYT 11140214)", sedimentological, geochemical and bathymetric studies offshore Lebu were performed (Fig. 1). In 2016 and 2017 two marine campaigns on board R/V Kay Kay II were carried out collecting bathymetric data, seawater samples and marine sediments.

Marine sediment samples were recovered using a gravity corer (diameter equal to 9 cm) at around 127 mbsl, and it drilled as deep as 240 cm into marine sediments (core GC-02). The core was collected at the northern of the positive relief (near 100 m) close to 73°44'25"W-37°36'10"S (Fig. 2) and then it was divided into four sections of 60 cm long (S01, S02, S03, S04). Each part of the core was frozen on board and later analysed at the Sedimentology Laboratory, University of Andres Bello (Viña del Mar, Chile).

The water samples were collected by Niskin bottles at five depths (0 m, 10 m, 20 m, 50 m and seafloor) and temperature, conductivity, dissolved oxygen and pH were determined with the multiparameter Meter model IP67. These parameters were measured at the two ends of the identified lineament, i.e., the first station located to the south and the second one to the north (Fig. 2). The measurements were obtained in the vicinity of relief to evaluate the relationships between marine sediments and the water column in presence of gas hydrate.

**2.2 Methods**

The procedure is based on a multi- and interdisciplinary approach to completely characterize the system, by using field and laboratory data, theory and modelling. The strategy includes: a) bathymetric data processing and b) sedimentological, physical-chemical, geochemical and biological analyses of seawater and marine sediment samples.

Bathymetric and sound velocity data were acquired using a multihaz Reson SeaBat 7125 echosounder (400 kHz, 0.5° x 1°), an SVP90 probe, and an AML Oceanographic Model Minos X sound velocity profiler. Preliminary processing was performed on board using a PDS2000 commercial software, which allows correcting bathymetric data in real time using the SVP90, AML information and ship motions (pitch, roll, yaw and heave). The bathymetric data processing was performed using the open-source MB-System software (Caress and Chayes, 2017). In this step, bathymetric data were converted in MB-System format to attenuate tide and scattering effects. In the first step, bathymetric grids were created with nearest neighbour interpolation algorithm, using the open-source software Generic Mapping Tools (GMT, Wessel et al., 2013). The algorithm builds GRID values in depth rectangular distributed, in which each node value corresponds to the weighted average of SAMPLES around a 5 m search circle. The selected grid was configured with a spatial resolution of 1 m. Finally, a median filter of 5 m width was applied to smooth the grid.

Grain size analysis includes sieving method where sediments pass (by agitation) through meshes; in our case, 50 g of sediment samples were sieved by using the following mesh sizes: 60, 80, 120 and 230. The pipette method was adopted in order to separate clay and silt fractions by selecting 15 g of mud sample. Statistical parameters were calculated in agreement with reported formulas (Folk and Ward, 1957; Carver, 1971; Scasso and Limarino, 1997).

Seawater physical-chemical properties (temperature, pH, salinity and dissolved oxygen) in the proximity of the positive relief were obtained using the multiparameter Meter (IP67, model 8602). The multiparameter Meter has different types of probes or electrodes, which must be selected according to the required function to obtain accurate measurements. The temperature was measured in Celsius degree, with an accuracy of ±0.5°C, while pH was directly related to the ratio of the concentrations of hydrogen ions [$H^+$] and hydroxyl [OH] (Cabo, 1978) with an accuracy of ±0.1. Salinity was obtained from the conductivity, which depends on the number of dissolved ions per unit volume and the mobility of the ions; the accuracy is ±0.1. Finally, dissolved oxygen can be measured both in % and in mg/L, with an accuracy of ±3%; in our case, it was expressed in %.

The core was cut in sections of 10 cm long, and then the main physical-chemical parameters were measured including water content (%), porosity (Φ), the content of solid material per unit volume, expressed as apparent density (ρ; Salamanca and Jara, 2003) and total organic carbon (TOC). Finally, the samples were dried in a forced air oven at 60°C for 36 hours and in a desiccator for 30 minutes.

TOC content was measured by gravimetric determination of weight loss through the
loss-on-ignition method (Byers et al., 1978; Luczak et al., 1997). In our case, 2 g of
dry sediment sample was calcined in a muffle at 500 °C for 5 hours and then it was
placed in a desiccator for 30 minutes until to register constant weight to reduce the
associated error.

For the foraminifera extraction, the corer was cut into sections of 15 cm from which
50 g of material was extracted, which was washed, dried and sieved, 120 and 230
sieves were used. The species were selected under binocular magnification, being
deposited in Petri dishes separated by the group for their later identification. The
general morphological features of the specimens were classified using the Atlas of
Benthic Foraminifera (Hobourn et al., 2013) and the genus was identified based on
the study of Chilean material (e.g. Figueroa et al., 2005).

Pore water from the core was extracted using an ACME lysimeter (0.2 µm) to
analyse oxygen and deuterium stable water isotopes. The pore water extraction
procedure includes: a) corer cutting in sections of 5 cm long; b) centrifugation; c)
pore water extraction by using Rhizon MOM with pore sizes ranging from 0.12 to
0.18 µm; and d) stable water isotope determination by Cavity Ring-Down
Spectroscopy (CRDS) method at the Laboratorio de Análisis Isotópico (LAI) at the
Universidad Andrés Bello (Viña del Mar, Chile).

Oxygen and deuterium water isotope analyses were evaluated using in-house
standards LIMS (Coplen and Wassenaar, 2015) and normalized to the VSMOW-
SLAP scale and reported as δ-values for oxygen ($\delta^{18}O$) and deuterium (δD). Each
sample was measured at least twice in different days. For each measurement,
samples were analysed for five consecutive times. Results are accepted if the
standard deviation of every single run (composed of five repetitions) is <1‰ for δD
and <0.1‰ for $\delta^{18}O$. Thereafter, the accepted stable water isotope value of a sample
is the average of the (at least) two different valid measurements within the range of
the previously explained standard deviation.

**3. Results**

From bathymetric data, a positive relief located at 127 mbsl with orientation N55°E
was recognised. The relief shows an average elevation of about 6 m above the
seafloor, an extension of 410 m length and a width of 50 m reaching an area of
14640 $m^2$ (Fig. 2). Five peaks ranging from 3 to 9 m high along the relief were
observed.

Grain size analysis shows constant values with depth. The average grain size
corresponds to sandy mud textural group. Silt-size reaches 60% of the total volume
(Fig. 3). Physical-chemical parameter distributions of core GC-02 are detailed in
Table 1. A slightly variation of water content (W) ranging from 38.6 to 46.3 %
(average equal to 43.1%), porosity (ɸ) ranging from 62.7 to 69.7 % (average equal
to 66.9%) and apparent density (ρ) ranging from 1.5 to 1.7 $g/cm^3$ (average equal to
1.6 $g/cm^3$) were detected. TOC values show a variable trend with a maximum value equal to 8.7% of total volume located at 2.2 m, while the minimum value is equal to
5.1% of total volume detected at 0.4 m (Fig. 4). Note, as expected, an opposite trend
distribution was recognized between porosity and apparent density.

Pore stable water isotope analysis of the marine sediment core shows positive
values ranging from 0.0 up to +1.8‰ for of δ$^{18}$O and 5.6‰ for δD, respectively (Fig
5). Stable water isotope δ-values show a positive trend (enrichment) towards the
bottom of the sediment core, with values close to 0 at the top in the sediment-
seawater interface, and a restricted variability for all samples analysed (Std. Dev.
0.33 and 0.95 for δ$^{18}$O and δD, respectively). It was noticed that no negative values
were found along the core.

Benthic foraminiferal accumulations were found in shallower levels of the core (0-60
cm) showing globose and elongated morphologies. The following genera and
species of opportunistic foraminifera were identified: *Globobulimina sp., Bolivina*
*plicata, Anomalinoides sp., Uvigerina peregrina, Oridorsalis tener* and
*Quinqueloculina vulgaris* (Fig. 6).

Respect to the properties of the water column, temperature range from 12 to 14 °C
in seawater samples, registering maximum values in correspondence of shallower
levels, while minimum values were found in deeper levels. Salinity and dissolved
oxygen show a similar trend with maximum values equal to 33‰ and 60% located
at 20 mbsl, respectively. Minimum values of salinity (31 ‰) and dissolved oxygen
(66.2%) were measured in station 1 (see Fig. 2 for location) at 0.6 mbsl. pH values
range from 7.5 to 8.1.

**4. Discussion and conclusion**

The stable water isotope composition of pore water represents strong evidence of
gas-hydrate dissociation. Figure 5a shows the stable water isotope profile of the
entire core, showing a clear increase with depth, with values close to 0‰ at the
seawater-sediment interface to positive values at the bottom of the core (1.8‰ for
δ$^{18}$O and 5.6‰ for δD). According to observational data for similar latitudes and
modelled surface water stable isotope composition for this ocean region, shallow
water should have a slightly negative isotope composition (~-0.2 to -0.5‰ δ$^{18}$O)
(Schmidt et al., 1999; LeGrande and Schmidt, 2006), which are related to the
transport of Subantarctic Waters through the Humboldt Current System along the
Chilean coast (Silva et al., 2009). The given negative values are mainly influenced
by the mix of oceanic and depleted meltwater from the Antarctic Ice Sheets (Sharp,
2007). Therefore, the observed trend shows the influence of seawater mixing on the
top and a different source at the bottom of the core. Positive values of meteoric
waters are mostly associated with high evaporation rates, which must be discarded
in the context of this investigation. Positive δ$^{18}$O values have been reported for clay
minerals dewatering; however, in this case, a δD depletion rather than enrichment
is expected (Hesse, 2003). Nonetheless, the co-isotope relationship (Fig. 5b) of our
samples shows that pore waters stable water composition have a strong ($r^2$=0.8)

positive correlation (i.e. simultaneous enrichment of $\delta^{18}$O and $\delta$D). Additionally, the
meteoric origin of the pore water can be rejected as shown in Fig. 5c, as pore waters
fall away from the Global Meteoric Water Line (GMWL), which defines the
fractionation processes during the hydrological cycle (Craig, 1961). Stable water
isotope enrichment of pore has been related to hydrate melting during coring and
post-sampling (Hesse, 2003; Tomaru et al., 2006), which are preferentially enriched
by heavy stable water isotope. Finally, these results are in agreement with values
reported by Rivera, (2018) for three sediment cores on this relief.

The infaunal foraminifera, found in the shallower sediment samples (e.g. Bolivina
Globobulimina and Uvigerina), could be associated with modern cold seeps, since
they can metabolize seeping methane, directly or indirectly exploiting the available
geochemical energy source (Jones, 2014). In fact, several authors reported these
genera over the continental margins of California, Japan and Mexico (Bernhard et
al., 2000; Rathburn et al., 2000; Hill et al., 2003), where benthic communities were
identified living over seepage areas. Besides, benthic foraminifera is associated with
high organic content ambient, low oxygen conditions and cold seep occurrences (Hill
et al., 2003; Rathburn et al. 2000).

In the study area across the continental slope zone, gas phases concentrations were
estimated by Vargas-Cordero et al. (2010a), reporting 15% of total volume for
hydrates and 0.2% of total volume for free gas. Several studies argue that lateral
fluid migration can occur from deep levels through faults and fractures canalising
fluids and giving place to mud mounds and mud volcanoes (e.g. Yin et al., 2003;
Thatcher et al., 2013). Other researches in our study area have reported faults
extending wards offshore zones, in particular, the Santa María fault shows a similar
orientation (N55°E) than the documented relief in this study (Melnick et al., 2009;
Vargas-Cordero et al., 2011; Becerra et al., 2013). Moreover, gas accumulations can
reach shallow areas because the base of gas hydrate stability zone (GHSZ) can be
very shallow in the continental shelf, as indicated by theoretical modelling. In fact, in
order to understand where the gas hydrate is stable versus seawater depth, the
theoretical base of the GHSZ was calculated assuming a geothermal gradient of 30
°C/km (in agreement with Vargas-Cordero et al., 2010a) and a mixture of 95% of
methane and 5% of ethane (in agreement with measures at ODP Site 1235, Mix et
al., 2003). The theoretical base of the GHSZ is calculated as the intersection
between the hydrate stability curve and the temperature/pressure curve in the
sediments (i.e., Tinivella and Giustiniani, 2013). The first curve is evaluated by using
the Sloan (1998) equations, which are used to model a mixture of gases in
freshwater. Then, Dickens and Quinby-Hunt (1994) equations are used to shift the
freshwater hydrate curve due to the effect of the water salinity, in our case equal to
3.5% (Vargas-Cordero et al., 2017). The second curve is evaluated considering the
water density equal to 1040 kg/m$^3$, in agreement with Vargas-Cordero et al. (2017).
It is crucial to notice that in our study area the presence of the hydrate and the free
gas has been detected by seismic data, confirming that this area is characterised by
relevant upward fluid flow (Vargas-Cordero et al., 2010a; Vargas-Cordero et al.,
2011). Fig. 7 reported the main geological features, including the seismic indicator of the transition between the gas hydrate and free gas zones, the so-called bottom
simulating reflector (BSR), detected in a seismic line located nearby the relief.

Fig. 8 shows the theoretical base of GHSZ reaches the seafloor at a seawater depth
of about 400 m; so, at shallower seawater depth the hydrate is not stable and only
free gas can be present. Note that in our study area the continental shelf is narrow
(15 km width) favouring that fluids associated with gas hydrate dissociation and gas
accumulations can migrate to shallow areas from the base of GHSZ. It is important
to notice that in other areas at higher latitudes, an extent reduction of the GHSZ, was
observed due to the warming over the last 20,000 years (i.e. Westbrook et al., 2009;
Thatcher et al., 2013). To verify a similar trend in our study area, we modelled the
theoretical base of the GHSZ supposing past temperature conditions reported by
paleoclimatic reconstruction studies (Kim et al., 2002; Lamy and Kayser, 2009), i.e.
a decrease of the seawater bottom temperature of 1 °C, 2 °C, 3 °C, 4 °C, and 5°C
(Fig. 8). The other parameters necessary to evaluate the hydrate stability
(geothermal gradient, water depth and gas composition) are supposed unvaried.
The modelling indicates that the origin of the mud structures analysed in this paper
can be probably related to hydrate dissociation caused by the increase of seawater
bottom temperature in the past, even if additional measurements should be
necessary to validate our hypothesis.

Grain size results can be associated with hydrodynamic conditions, in which mud
and sand could be related to coastal and beach systems, fluvial or deltaic deposits
(Mordojovich, 1981). Slightly vertical variations allow us to define a relationship
between physical-chemical parameters (W, $\phi$, $\rho$ and TOC) and grain sizes results.
Studies reported by Pineda (2009) argue that clay and silt presence in marine
sediments are capable of retaining retain organic wastes increasing TOC values.
The values ranging from 0.5 to 10% reported by Pineda (2009) are in agreement
with the values presented in this study. Then, TOC values, sediment types and grain
sizes confirm that the geological context is compatible with the gas hydrate formation
(i.e., White et al., 2009; Hyodo et al., 2013).

The results from seawater analysis show typical values of temperature, salinity,
dissolved oxygen and pH, which are associated with seawater masses. The
temperature in the seawater column increases in shallow levels, whereas it
decreases in deep levels. An opposite trend regarding salinity and dissolved oxygen
values were recognized; in effect, when the oxygen solubility decreases, the
temperature and salinity increases (Cabo, 1978). pH values ranging from 7.4 to 8.4
can be associated with seawater alkalinity. Higher values often are detected on the
seawater surface (Cabo, 1978). No relationships were found between seawater
physical-chemical parameters and our conclusion, which can be explained due to:
a) discrete data collected (e.g. five seawater samples were collected in a column of
130 m) or b) upwelling and downwelling processes reported in this area (Parada et
al., 2012) could give place to water mass exchange preventing to observe significant
variations.

In conclusion, our results indicate that the positive relief could be associated with
mud mound growing by fluid flux supply, which could be canalised through faults and fractures, predicted by our analysis and detected by seismic data (Vargas Cordero et al., 2011). Moreover, $\delta^{18}O$ and $\delta D$ enrichment of pore water, related to gas hydrate melting and dissociation, actively support this observation. Additionally, Bolivina, Uvigerina and Globobulimina genera found in shallow sediments support our conclusions. Based on our static modelling, these fluids related to gas hydrates dissociation (a mix of freshwater, mud and gas) might migrate from deeper to shallower depths, reaching shallow sediments and giving place to mud mound growing.

**Acknowledgements**

We are grateful to CONICYT (Fondecyt de Iniciación N°11140216), which partially supported this work. The authors are grateful to Michela Giustiniani for constructive discussions and useful comments. Special thanks to Mauricio and Daniel from the palaeontology laboratory (UNAB - Viña del Mar), who helped us with the foraminifera identification.

**Figures**

[Figure]

**Figure 1**: Location map of the studied area. Red star shows core recovery and
bathymetric survey. The dashed line shows the bathymetric profile used in Fig. 8.

[Figure]

**Figure 2**: Bathymetric map indicating location core GC-02 (red circle). In A) and B)
3D images with orientation NW and SW respectively. The white circles indicate the
position of the two water samples.

[Figure]

**Figure 3**: Grain size distribution in marine sediments (core GC02).

[Figure]

**Figure 4**: Physical-chemical parameters distribution in marine sediments (core
GC02).

[Figure]

**Figure 5**. Oxygen (δ¹⁸O) and deuterium (δD) stable water isotope distribution in sediment from: a. Depth profile of the core, b. co-isotope linear regression of pore water samples and c. Co-isotope relationship of pore water samples against the global meteoric water (GMWL).

[Figure]

**Figure 6**: Benthic foraminifera. In (1a) *Globobulimina sp.*, lateral view (10x); (1b)
*Globobulimina sp.*, lateral view (10x); (2) *Bolivina plicata.*, lateral view (5x); (3)
*Anomalinoides spp.*, lateral view (5x); (4) *Uvigerina peregrina*, lateral view (5x); (5a)
*Oridorsalis tener*, lateral view (5x); (5b) *Oridorsalis tener*, lateral view (5x); (6)
*Quinqueloculina vulgaris*, lateral view (10x).

[Figure]

**Figure 7**: Line drawing section corresponding to the SO161-44 seismic stacking
section modified from Fig. 3 in Vargas-Cordero et al. (2011), in which the main
geological features and the seismic reflector indicating the transition between gas
hydrate and free gas (i.e., the bottom simulating reflector, BSR) are indicated. In the
inset, the location map showing static model location reported in Fig. 8 (red dashed
line), and mud growing zone (red circle) and the SO161-44 position (black line).

[Figure]

**Figure 8**: Schematic profile explaining mud growing formation (in red). The profile
location is shown in Fig. 1. Dashed lines show theoretical bases of GHSZ by using
the geothermal gradient of 30°C/km for several scenarios supposing that the hydrate
is formed by a mixture of 95% of methane and 5% of ethane. The blue dashed line
indicates the actual theoretical base of the GHSZ. The dotted lines indicate the
theoretical base of GHSZ supposing a decrease of the bottom temperature of 2 °C
(black dotted line), 3 °C (magenta dotted line), 4 °C (green dotted line) and 5 °C (red
dotted line). The black solid line indicates the seafloor. The pink arrows indicate the
direction of the fluid/mud outflow. Possible faults and fractures are also reported as
black lines.

| Depth (m) | W (%) | φ (%) | ρ (g/cm3) | TOC % |
|---|---|---|---|---|
| 0.1 | 45.2 | 68.8 | 1.56 | 7.9 |
| 0.2 | 42.2 | 66.1 | 1.61 | 7.2 |
| 0.3 | 41.2 | 65.2 | 1.62 | 6.6 |
| 0.4 | 41.3 | 65.3 | 1.62 | 5.1 |
| 0.5 | 39.9 | 64.0 | 1.64 | 5.9 |
| 0.6 | 38.6 | 62.7 | 1.67 | 6.0 |
| 0.7 | 40.3 | 64.4 | 1.64 | 5.9 |
| 0.8 | 42.7 | 66.5 | 1.60 | 6.4 |
| 0.9 | 43.0 | 66.8 | 1.60 | 6.2 |
| 1 | 42.8 | 66.6 | 1.60 | 6.5 |
| 1.1 | 42.6 | 66.5 | 1.60 | 6.6 |
| 1.2 | 42.9 | 66.7 | 1.60 | 6.1 |
| 1.3 | 45.0 | 68.6 | 1.57 | 6.0 |
| 1.4 | 45.2 | 68.8 | 1.56 | 6.7 |
| 1.5 | 45.1 | 68.7 | 1.56 | 4.2 |
| 1.6 | 46.3 | 69.7 | 1.55 | 7.5 |
| 1.7 | 44.1 | 67.8 | 1.58 | 6.7 |
| 1.8 | 45.3 | 68.8 | 1.56 | 7.1 |
| 1.9 | 40.7 | 64.7 | 1.63 | 5.4 |
| 2 | 43.7 | 67.5 | 1.58 | 7.0 |
| 2.1 | 44.3 | 68.0 | 1.58 | 6.8 |
| 2.2 | 43.3 | 67.1 | 1.59 | 8.7 |
| 2.3 | 45.4 | 68.9 | 1.56 | 6.9 |
| 2.4 | 44.4 | 68.0 | 1.58 | 6.5 |
| **Average** | **43.1** | **66.9** | **1.59** | **6.5** |
| **Minimum** | **38.6** | **62.7** | **1.55** | **4.2** |
| **Maximum** | **46.3** | **69.7** | **1.67** | **8.7** |

**Table 1**: Physical-chemical parameter distribution in marine sediments.